# High-Yield Mucosal Olfactory Ensheathing Cells Restore Loss of Function in Rat Dorsal Root Injury

**DOI:** 10.3390/cells10051186

**Published:** 2021-05-12

**Authors:** Kamile Minkelyte, Andrew Collins, Modinat Liadi, Ahmed Ibrahim, Daqing Li, Ying Li

**Affiliations:** 1Spinal Repair Unit, Department of Brain Repair and Rehabilitation, UCL Institute of Neurology, Queen Square, London WC1N 3BG, UK; kamile.minkelyte.18@ucl.ac.uk (K.M.); 1andrewcollins@gmail.com (A.C.); m.liadi@ucl.ac.uk (M.L.); ahmed.ibrahim@ucl.ac.uk (A.I.); daqing.li@ucl.ac.uk (D.L.); 2Barking, Havering and Redbridge University Hospitals, London RM7 0AG, UK

**Keywords:** spinal roots, translational model, mucosal OECs, transplantation, spinal cord, rhizotomy, CNS repair

## Abstract

In a previous study, we reported that no axons were crossing from the severed dorsal roots to the spinal cord using the rat dorsal rhizotomy paradigm. The injury caused ipsilateral deficits of forepaw function. An attempt to restore the function by transplanting cells containing 5% olfactory ensheathing cells (OECs) cultured from the olfactory mucosa did not succeed. However, obtaining OECs from the olfactory mucosa has an advantage for clinical application. In the present study, we used the same rhizotomy paradigm, but rats with an injury received cells from a modified mucosal culture containing around 20% OECs mixed in collagen. The forelimb proprioception assessment showed that 80% of the rats receiving the transplants had functional improvement over six weeks of the study. The adhesive removal test showed that the time taken for the rats to notice the adhesive label and remove it almost returned to the normal level after receiving the transplants. Transplanted cells were identified with the expression of green fluorescent protein (ZsGreen). Some regeneration fibres immunostained for neurofilament (NF) or traced by biotinylated dextran amine (BDA) in the injury area were associated with the transplanted cells. The evidence in this study improves the prospect of clinical application using OECs from the olfactory mucosa to treat CNS injuries.

## 1. Introduction

It is known that the olfactory system is the only part of the mammalian central nervous system where axon regeneration continues throughout adult life [1,2]. Olfactory ensheathing cells (OECs) are considered to play a significant part in this process. Transplantation of OECs into the lesioned CNS region has been shown to induce axonal regeneration and restore lost functions [3,4,5,6,7,8,9,10,11,12,13,14,15,16,17].

OECs can be obtained from the olfactory bulb and the olfactory mucosa [6,18,19,20,21,22]. With a view of developing a method for application in a clinical setting, it would be preferable to avoid the necessity for craniotomy and bulbectomy to retrieve the olfactory bulb and instead obtain OECs for transplantation by the less invasive intranasal approach of a biopsy from the olfactory mucosa [13,23,24,25,26,27]. However, clinical trials of transplantation of OECs into spinal cord injuries using tissue fragments or cells cultured from the olfactory mucosa showed little or no neurological improvement [24,28,29,30]. Our previous study of transplanting cultured cells derived from the olfactory mucosa which had less than 5% of OECs in a rat dorsal root injury model showed no functional restoration of forepaw grasping [31].

We have now modified our previous protocol to culture OECs from a mucosal tissue sample and increased the OEC yield by fourfold, reaching up to a 20% proportion compared to 5% previously. Our lab has already developed a method of encapsulating cells in collagen gel to maximise the limited number of cells from biopsied human olfactory bulb tissue samples [32]. The fabrication of a biomaterial scaffold for cell transplantation provides the possibility to bridge lesions of considerable sizes, such as in spinal cord injuries, stabilise the cells in the graft location, and improve the surgical manoeuvrability of the transplant [33,34]. The use of biomaterials can also facilitate the integration of transplanted cells with the host tissue [35]. In the present study, we report that the forelimb proprioception could be restored in a vertical climbing task by transplantation of a high yield of mucosal OECs encapsulated in collagen gel. This functional improvement was not observed in the rats receiving around 5% of the OEC population with our previous protocol. There was also a marked improvement in the adhesive removal function test. The time taken to notice and remove the adhesive label almost returned to the normal level in the rats receiving transplants of high yield of mucosal OECs.

## 2. Materials and Methods

In this study all animals were used in accordance with the UK Home Office regulations for the care and use of laboratory animals, the UK Animals (Scientific Procedures) Act 1986, with the ethical approval of the University College London Institute of Neurology. Adult male Sprague–Dawley rats (Body weight: 150–200 g; SD) were purchased from Charles River, UK. Thirty-five animals were used for in vivo experiments and 10 for cell culture.

### 2.1. Preparation of High-Yield OECs from the Olfactory Mucosa (OM)

Rats of 250 g (*n* = 10) were culled by exposure to a rising carbon dioxide (CO_2_) concentration followed by decapitation. Under sterile conditions, the nasal septum was exposed. Under a dissecting microscope, the olfactory mucosa was identified for its amber colour and dissected from each side of the septum. OM tissue pieces were collected into ice-cold DMEM/F12 + penicillin/streptomycin (100 U/mL/100 µg/mL; PS). After excessive mucus was removed, they were transferred into a 35 mm petri dish with fresh medium + PS and cut into small pieces followed by incubation in DMEM/F12 containing 25 mg/mL collagenase (Type I, Sigma-Aldrich, Gillingham SP8 4XT, UK) at 37 °C for 45 min. The enzymatic reaction was stopped with the addition of Hanks’ Balanced Salt Solution (HBSS). The tissue was dissociated by gentle trituration with a 1 mL plastic pipette tip and then filtered through a 0.45 µm cell strainer into a 50 mL falcon tube generating a single-cell suspension. The cell suspension was centrifuged at 300× *g* for 5 min and the cells were resuspended in a fresh medium after the supernatant was removed. The cells were counted on a cell counter (Countess, C10228, Thermo Fisher Scientific, Dartford DA2 6PT, UK) and seeded at 2.6 × 10^6^ on 35 mm dishes coated with poly-l-lysine (PLL, Sigma-Aldrich, UK). The cells were fed every second day with a medium composed of DMEM/12 + 10% foetal bovine serum (FBS) + 1% mixture of insulin-transferrin-selenium (ITS) + 0.02% 10 µM Forskolin (FSK) + 1% N-2 100X Supplement + PS and cultured for 21 days when they became confluent (all media and reagents were purchased from Thermo Fisher Scientific, UK, unless specified).

### 2.2. Lentiviral Transduction of ZsGreen Fluorescent Protein

On the 20th day, the cultured cells were transduced to express ZsGreen fluorescent protein. The cells were exposed to lentiviral vectors rLV.EF1.ZsGreen1-9 (Takara Bio Europe, Saint-Germain-en-Laye, France) for 12 h in a fresh medium. After the virus-containing medium was discarded, the cells were washed thoroughly with DMEM/F12 to remove remaining the viral particles. The cells were cultured for a further 2 days in collagen gel before transplantation.

### 2.3. Encapsulation of Modified Mucosal OECs (mMOECs) in Collagen

The details of cell/collagen preparation have been described in our previous publication [32]. The cells were incubated in trypsin/EDTA (TE) to be lifted from the bottom of culture dishes. The activity of TE was inactivated by adding DMEM/F12 + 10% FBS, and the cells were collected and triturated into a single-cell suspension. Encapsulation of the cells in collagen was accomplished by mixing a cell suspension with type 1 collagen (SLS, Nottingham NG11 7EP, UK), 1M NaOH, and Modified Eagle’s Medium (MEM, 10×, Sigma-Aldrich, UK). A volume of 250 µL of cell/collagen mixture was deposited to a 35 mm culture dish and placed into an incubator where it polymerised to form a gel matrix. Once the gel became firm it was submerged in a culture medium until use. The medium was changed every other day as was the cell culture. Each collagen gel contained approximately 2 × 10^5^ cells with a diameter of around 1 cm and a thickness of around 1 mm. The gel was trimmed to approximately 2 mm^2^ pieces before they were transplanted.

### 2.4. Surgery

#### 2.4.1. Control—Dorsal Rhizotomy and Sham Surgery (*n* = 14)

Unilateral transection of four dorsal roots was carried out as described in our previous study [32]. Briefly, under isoflurane anaesthesia, a skin incision was made along the dorsal midline, the muscles were separated, and the prominent T2 process was located. Hemilaminectomies were performed from C4 to T2, and the dura was incised with a pair of microscissors to reveal the dorsal roots. The roots of C6, C7, C8, and T1 were transected with microscissors as close as possible to the spinal cord in a plane approximately perpendicular to their entry into the spinal cord. The cut roots were reapposed and held in place with fibrin glue (Tisseel Kit, Baxter, Thetford, UK). In a pilot study, four rats received a sham surgery performance by exposing the four unilateral dorsal roots and the spinal cord, but without cutting the nerves.

#### 2.4.2. Transplantation—Rhizotomy with Transplantation of mMOECs (*n* = 21)

Twenty-one rhizotomised rats received transplants of modified MOECs within a collagen scaffold after unilateral dorsal roots of C6, C7, C8, and T1 were transected as above. The transplant was applied between the cut ends of the roots and their original entry point on the spinal cord and held in place with fibrin glue (Tisseel Kit, Baxter, Thetford, UK).

#### 2.4.3. Biotinylated Dextran Amine (BDA) Labelling (*n* = 11)

Details of the procedure were described in our previous publication [16]. Briefly, seven rhizotomised rats with transplanted mMOEC were used for tracing regenerating DR fibres by injection of biotinylated dextran amine (BDA, Thermo Fischer Scientific, UK) into the ganglia 4 weeks after transplants. This was also conducted in rhizotomy alone animals (*n* = 4). In addition to C6–T1, the C5 and T2 adjacent roots were also sectioned to avoid a tracer passing through anastomoses to adjacent intact roots. About 1.5 µL of 10% BDA in saline was injected by a glass micropipette at three points into each of the two middle (C7 and C8) DR ganglia. The rats were terminated and perfused two weeks after receiving the tracer injection.

#### 2.4.4. Postoperative Care

After dorsal root transection or transplantation, the overlying paraspinous muscles and skin were subsequently sutured in layers (Vicryl Plus, Ethicon, VWR, Leicestershire LE17 4XN, UK). The animals were placed in a warm cage to recover from anaesthesia before returning to their home cage. They were given a wet diet and postoperative pain relief (0.05 mg/kg buprenorphine, s.c. daily for 3 days). Bitter paste (Henry Schein) was applied to the forepaw on the operated side every other day for the first 2 weeks to prevent autotomy. Autotomy was observed in one animal in the control and one in the transplanted group. If the autotomy occurred to the extent of involving two joints in one toe, or past the nail, in two toes, the animals were culled humanely (under the guidance of the Project License). The rat housing room was maintained between 20 °C and 22 °C under standard lighting conditions (12:12 light–dark cycle) with food and water available ad libitum.

### 2.5. Function Analysis

#### 2.5.1. Vertical Climbing Test

The rats were handled daily and placed on a 1-metre near-vertical grid (15° inclination) to climb up the rungs to the top 2 weeks before surgery and for up to 6 weeks post-surgery. The rats were filmed climbing up the vertical grid once a week. In the subsequent slow-motion analysis, the rat forepaw on the injured side was scored for a degree of accuracy in locating and grasping the grid bars as described in a previous study [16,32]. Briefly, the grasp was recorded as successful when there was purposeful movement ending in full functional flexion of the digits to grasp the bars. If a rat grasped the bars successfully with all attempts for one complete climb it was scored as 0% error; if it failed to grasp the bars with all attempts, it was scored as 100% error. Unsuccessful grasps were graded from 1 to 4 in increasing order of severity depending on how far the forelimb protruded through the grid ((1), paw reaches grid level but no grasping; (2), protrudes through the grid as far as the wrist; (3), as far as the elbow; (4), as far as the axilla. Figure 1). The assessments were carried out blindly by two people for experimental groups, and the scores from the two assessors were averaged.

#### 2.5.2. Adhesive Removal Test

A total of 14 rats were used for the adhesive removal test. Six rats were randomly selected to have rhizotomy alone to serve as the control. The other 8 rats had rhizotomy with mMOEC transplants. The rats were placed into an empty Perspex box and allowed to acclimatise to their environment. A round 8 mm adhesive label was stuck on the middle of the palm of one of the forepaws. The rat was released, and the length of time taken for the rat to notice the adhesive label (with purposeful intent to remove it) was recorded. Once the adhesive label was taken off, the same procedure was repeated on the other forepaw. If the adhesive label was not removed by 180 s, the test was stopped, and that time was recorded. The first paw tested between the left and right sides was randomised to prevent conditioning. Four repeats were carried out per alternating forepaw in one session, and the rats were tested twice a week for 6 weeks.

### 2.6. Immunohistochemistry

#### 2.6.1. Cell Culture Evaluation

After the cells were cultured for 21 days, they were fixed for 30 min with 4% paraformaldehyde in 0.1 M phosphate buffer (4% PFA, Sigma-Aldrich, UK), washed in 0.01 M phosphate-buffered saline (PBS), and incubated in a blocking solution containing 2% skimmed milk (Oxoid LP6031) + 1% Triton X-100 (Fisher Bioreagents, BP151-500, Fairlawn, NJ, USA) for 60 min. The cells were incubated overnight at 4 °C with a cocktail of primary antibodies: mouse monoclonal antibody against low-affinity nerve growth factor receptor (P75, 1:500, Millipore, Temecula, CA, USA) and rabbit polyclonal antihuman fibronectin (FN, 1:1000, Dako, Denmark). After washing in PBS, the cells were incubated with fluorophore-conjugated secondary antibodies (goat antimouse Alexa 488, and goat antirabbit Alexa 546, both 1:500, Molecular Probes, Invitrogen) for an hour, washed in PBS, counterstained, and mounted using Prolong Gold antifade reagent containing the nuclear dye 40, 6-diamidino-2-phenylindole (DAPI, 0.4 μg/mL, Thermo Fisher Scientific, UK). The culture area was divided into 4 quadrants on each culture dish, and four images sized 1.02 mm × 0.76 mm in each quadrant through each of the three fluorescent channels (red, blue, and green) were captured on a Nikon Eclipse 55i microscope at 100× magnification. The total cell number and the subpopulation of OECs and olfactory nerve fibroblasts (ONFs) in the sampled area were counted with the Fiji software (based on Image J developed by NIH).

#### 2.6.2. For In Vivo Study

Six weeks after surgery, after deep carbon dioxide anaesthesia, the rats were transcardially perfused with 50 mL PBS followed by 400 mL 4% PFA for 30 min. The vertebral columns were dissected from the craniocervical junction to the upper thoracic level and left to harden in the same fixative for 2–3 days at 4 °C. The spinal cord and associated roots were carefully dissected under a dissecting microscope to preserve the continuity across the dorsal roots and transplants to the spinal cord. The tissues were placed sequentially into 10 and 20% sucrose solution until they sunk. The tissue block was mounted with an embedding compound Cryo-M-Bed (Fisher Scientific, UK) on a cutting chuck and frozen in the Cryostat chamber (Leica CM3050) at −20 °C until hardened. Serial sections of a 16 µm horizontal plane at cervical levels C6–T1 were cut and thaw mounted onto slides. For immunohistochemistry, all sections were incubated in a 2% milk PBS blocking solution before applying antibodies diluted in 2% milk PBS. Between each step, the sections were washed in PBS (3 × 10 min washes). For double immunostaining, sections were incubated in the mouse monoclonal antibody against glial fibrillary acidic protein (GFAP, 1:1000) and the rabbit polyclonal antibody against laminin (LN, 1:500, both from Sigma-Aldrich, UK) overnight at 4 °C. The secondary antibodies used were the Alexa Fluor 546 and 488 antimouse and antirabbit (1:400) for 1–2 h in the dark at room temperature. For neurofilament (NF) immunostaining, sections were incubated overnight with 1:500 rabbit polyclonal antibody against neurofilament 68 kD/200 kD (1:500, Abcam, Cambridge CB2 0AX, UK) followed by 1:1000 Alexa Fluor 546 antirabbit secondary antibody. To enhance the fluorescent intensity of expression of ZsGreen, the antibody against ZsGreen (1:500, Living Colors^®^ Anti-RCFP Polyclonal Pan Antibody, Takara Bio Europe) and the Alexa 488 secondary antibody were used. To detect BDA labelling, sections were incubated in streptavidin conjugated with Alexa Fluor 546 (1:400, Thermo Fischer, Dartford DA2 6PT, UK) overnight at 4 °C. Fluorescent images were visualised and captured using a TCS SP1 Leica confocal microscope. The laser intensity, pinhole sizes, and sensitivity of the photomultipliers were adjusted accordingly to ensure no bleed through to cross channels.

### 2.7. Statistical Analysis

Results are expressed as means ± SEM, with the statistical comparison between groups using a two-way analysis of variance (ANOVA) to determine significance. Post hoc analysis was performed with Sidak’s multiple comparisons, and GraphPad Prism 8.0.1 software was used. Details of animal numbers are given in Materials and Methods, Results, and figure legends.

## 3. Results

### 3.1. Modified Mucosal OECs Culture and Encapsulation in Collagen

There are two major types of cells OECs and ONFs in the culture identified by immunofluorescence staining for P75 and fibronectin (FN). Under the previous culture condition, ONFs proliferated predominantly at a high rate and quickly occupied a large area of the culture dish. OECs were comparatively quiescent in proliferation. They either formed small clusters or resided individually within some narrow space spared by ONFs. At an early stage of culture around one week, some OECs differentiated into spindle-shaped morphology with small cell bodies, narrow cytoplasm, and bipolar thin and elongated processes.

As the culture progressed, the dominating expansion of ONFs led to the morphology of the OECs changing into large and flat shapes with irregular shapes and short processes appearing senescent. With the new protocol, more OEC colonies could be seen in the culture; it appeared that the rate of OEC proliferation increased. They maintained their spindle shape with thin cell bodies, narrow cytoplasm, and fine and long processes (green in Figure 2A). Many processes ran along with each other in parallel. The analysis of fluorescent micrographs showed the OEC proportion was increased up to 20% from 5% with the previous protocol. Figure 2A is a representative image to show that OECs (P75, green) occupy a large area. 

We optimised parameters to fabricate mMOECs encapsulation in collagen. We mixed the collagen at a concentration of 4–4.8 mg/mL and cells at a density of 8 × 10^5^/mL. The cells were distributed evenly within the collagen and formed a cellular meshwork (Figure 2B,C). Immunofluorescence staining showed the amount of FN deposited by ONFs was reduced inside the collagen gel. The gel provided structural support for the cells and could be trimmed to suit the actual size and contour between the cut ends of the dorsal root and the spinal cord.

### 3.2. Vertical Climbing Test

The unoperated rats climbed the 1 m vertical grid at 15° inclination with a regular sequence of alternating fore- and hindpaw movements and grasping the bars of the grid to provide support for upward climbing. The average time taken to complete one climb ranged from 10 to 30 s. Sometimes a rat was reluctant to climb or paused in the middle of the climb. We did not take this as a measure because the time taken was so variable.

#### 3.2.1. Rhizotomy Alone and Sham Surgery (Control Group; *n* = 14)

After unilateral transection of the C6–T1 dorsal roots, the rats were able to climb, but the ipsilateral forelimb rarely managed to reach the bars of the vertical grid, and when it did, the digits did not grasp the bars. Analysis of the recorded film showed that before surgery, rats climbed the vertical grid with baseline error scores of 0 in both forepaws. One week following the transection of the C6–T1 dorsal roots, the rats had an error score of 100% in the ipsilateral forepaw on the operated side—every attempt to grasp a bar failed. The failure to locate or to grasp the bars persisted and did not improve over the 6 weeks of the study course (Figure 3, red circles). The mean error score was maintained at 100% throughout. Grasping was unaffected on the forepaw of the unoperated side, and the rats remained able to climb the grid. The rats with the sham surgery did not show any abnormal behaviour or functional deficits in this test after they recovered to their normal cage activities after the surgery.

#### 3.2.2. Rhizotomy with Transplantation of mMOECs (*n* = 21)

The rhizotomised rats with mMOEC transplants showed a progressive improvement in locating and grasping the bars, although they did not recover functional performance to the normal level. The errors of climbing performance in the ipsilateral forepaw on the operated side were scored around 80.5% ± 6.6 at 1 week following the unilateral transection of the C6–T1 dorsal roots and transplantation of mMOECs compared to 100% in the control group (Figure 3, green squares). The error score was reduced to 60.5% ± 8.5 after 2 weeks post-surgery. The performance improved progressively, and the error score reduced further to 30.74% ± 5.5 at 6 weeks post-surgery.

### 3.3. Forepaw Fault Score

When climbing an upright grid, normal rats accurately placed each of the four paws in regular sequence, with practically no faults locating the bars, following which they closed the digits in a grasp.

#### 3.3.1. Rhizotomy Alone (Control Group; *n* = 6)

After unilateral transection of the C6–T1 dorsal roots, there was a significant deficit in the ipsilateral forepaw in locating the grid bars, and grasping was almost totally abolished. The magnitude of error for each of the misplaced grasps was assessed. Misplaced grasps were graded from ‘0’ to ‘4’ (see Figure 1), with ‘4’ representing the worst level of fault in which the forelimb slid straight through to the axilla. Before the injury, rats had a forepaw fault score close to 0, where almost all grasps successfully located the bars of the grid during upward climbing movements. One week following the transection of the C6–T1 dorsal roots, the rats scored 3.8 ± 0.2 in the forepaw on the operated side (Figure 4, red circles). The forepaw on the unoperated side was unaffected, and the rats remained able to locate and grasp the bars of the vertical grid.

#### 3.3.2. Rhizotomy with Transplantation of mMOECs (*n* = 21)

The rhizotomised rats with mMOEC transplant progressively reduced the faults of locating and grasping the bars of the grid. They had a lower score of 2.9 ± 0.3 in the forepaw on the operated side at 1 week post-surgery than those in the control group. The fault score decreased further at 3 weeks post-surgery, down from 2.9 to 1.43, with the forearm sliding less far through the grid bars, to an area between the digits and the wrist (Figure 4, green squares). In comparison to the foot fault score at this point, the control group was about 4.0. From 4 weeks post-surgery, the fault score continued decreasing and reached 0.9 ± 0.1 at 6 weeks, meaning the misplacement of the forearm slide only to the palmar area of the forepaw. 

### 3.4. Adhesive Removal Test (n = 14) 

The length of time taken for the rats to notice the adhesive label and remove it was recorded in all groups pre-surgery and once a week post-surgery for 6 weeks. The normal rats took less than 5 s to notice the label and remove it (Figure 5A). There was no statistical difference between both right and left forepaws before surgery. The time taken to notice the label and remove it in the rats with rhizotomy alone (*n* = 6) on the ipsilateral forepaw of the operated side increased to around 100 ± 13.1 s (mean ± SEM) 1 week after surgery (Figure 5B), whereas it was less than 5 ± 1.2 s before surgery. The length of time persisted at about 70 ± 12.1 s over the 6-week testing period with no statistical difference between weeks 1 and 6. Time taken was unaffected on the contralateral forepaw of the unoperated side and remained unchanged around 5 s over a 6-week testing period. In the rats with rhizotomy and transplant (*n* = 6), the time taken for the task on the ipsilateral forepaw of the operated side was initially lengthened up to more than 85 ± 32.2 s 1 week after surgery, which was similar to the observation in the rhizotomy alone group. The time taken was progressively reduced to around 39.65 ± 20.44 at 3 weeks, 11.86 ± 2.74 at 5 weeks, and maintained at this level at 6 weeks, with no significant difference from the pre-surgery level (Figure 5C). The rats with the sham surgery did not show any abnormal behaviour or functional deficits in this test.

### 3.5. mMOECs Transplant

The transplanted cells were identified individually or as a concentrated group by their expression of ZsGreen protein or immunofluorescence staining for ZsGreen. The transplanted mMOECs in collagen gel remained at the transplanted site bridging the severed dorsal roots and the spinal cord (Figure 6A, asterisk). Some of them were found in a large cluster and remained within the collagen gel, as observed in human bulb OEC transplants [30]. Many cells migrated out from the collagen gel (Figure 6A,B, arrows), which often had elongated processes. Some of them migrated into the spinal cord, as we observed previously with transplanted bOECs. Once in the spinal cord, the cells were running parallel to the host dorsal column axons.

### 3.6. Immunostaining of Neurofilament (NF) and Biotinylated Dextran Amine (BDA) Tracing

The immunostaining for NF revealed that some NF-positive axons were present at the dorsal root entry zone (Figure 6A) in the rhizotomised rats with transplants. The regenerating DR axons crossed the bridge formed by the transplants and entered the spinal cord. Some NF positive and BDA labelled axons were ensheathed by the transplanted mMOECs (Figure 6A,B,E) as shown in our previous study with bOEC transplants. There were no axons of the severed dorsal roots crossing the entry zone or growing into the spinal cord in the control group of rats that had rhizotomy alone (Figure 6D). 

BDA tracing confirmed that the regenerating DR axons crossed the dorsal root entry zone through the transplant and grew into the spinal cord in the rhizotomised rats with transplants. Some labelled axons were ensheathed in a one-to-one fashion by elongated OEC processes comparable to our previous observation with the bOEC transplant studies. The regenerating axons arborised into the dorsal horn and some ascended in the dorsal columns after entering the astrocytic territory of the spinal cord (Figure 6E).

### 3.7. Astrocyte Responses at the Dorsal Root Entry Zone

The interaction pattern at the entry zone between the peripheral and central tissues induced by the mMOEC transplants was markedly different from that observed in the rhizotomy alone group. Double immunofluorescence staining of GFAP/LN as shown in Figure 6C identified the PNS tissue by the red fluorescence of LN and the CNS tissue green fluorescence of GFAP. The overlapped area where the PNS and CNS tissue mingle appeared yellow (Figure 6C). The astrocytic processes (Figure 6D) did not extend into the dorsal root entry zone (DREZ) in the rhizotomy alone group.

In horizontal sections, multiple strands of filamentous GFAP-positive astrocytes were seen to extend their processes for up to around 200 µm at the same angle towards the transplanted cells (Figure 6C), as observed in our previous studies with bOECs. The transplanted cells were orientated, extended out from the cut stump of the dorsal root toward the spinal cord, and weaved with a massive outgrowth of astrocytes from the spinal cord. The merge of the astrocytic processes, the transplanted cells, and the PNS tissue formed a bridge between the severed dorsal roots and the spinal cord.

## 4. Discussion

In the present study, we report that up to a 20% population of olfactory ensheathing cells (OEC) from rat mucosal culture can be achieved by a simple and reliable method. When 5% of mucosal OECs transplanted into a rat rhizotomy model previously, we did not observe regeneration of dorsal root axons crossing the dorsal root entry zone into the spinal cord nor any function restoration. Using the same paradigm, the present study demonstrated that our high-yield mMOECs can induce axonal regeneration and restore loss of functions. The recovering rates of the two functional tests appear to be similar. Although the dorsal rhizotomy used in the present study does not reflect the larger, more variable damage encountered in the clinical relevance of human spinal plexus avulsions or spinal cord injuries, it keeps the animals in a good condition but with persistent deficits in our functional tests, which have allowed us to undertake a series of OEC transplantation studies [16,31,32,36,37].

### 4.1. Cell Sources

OECs can be derived from the olfactory bulb and the olfactory mucosa [38,39]. Most experimental transplantation studies, especially those with a positive outcome, have used OECs of bulbar origin (bOECs) in experimental injury models [5,6,9,10,11,12,13,14,15,16,40,41] and clinical applications [30,42,43,44]. In our previous studies comparing the transplantation of OECs from the bulb and mucosa into corticospinal tract lesions, only bOECs induced axonal regeneration across the lesion [45,46]. There could be several reasons for the differences: (1) the inherent differences between OECs from the two sources [47,48,49]; (2) the proportion of OECs are far richer in the bulbar culture than the mucosal cultures [26,50]; and (3) the nature of accompanying cells. Astroglia and central neurons are present in the primary bulbar tissue samples. In contrast, the mucosal tissue samples include epithelial and glandular cells and mesenchymal stem cells [51]. Of the various factors, the most obvious difference, and in our view most important, is in the proportion of OEC yield in the culture. The culture of olfactory bulb origin consistently produces a higher proportion of OECs than the olfactory mucosa using a standard culture protocol.

However, a craniotomy would need to retrieve the olfactory bulbs [52]. This surgical procedure has inherent risks, such as bleeding, meningitis, or stroke. The olfactory mucosa biopsy tissue can be obtained by an endonasal approach, which is far less invasive and considerably safer [21,26,27,30]. Several clinical studies have demonstrated the safety in autologous mucosal OEC (mOECs) transplantation in human spinal cord injuries [21,30]. However, experiments and clinical trials of transplanting mOECs have shown little or no effective anatomical repair nor any effective restoration of function in experimental models and treated patients [21,30,31,53,54,55,56]. 

### 4.2. Culture Protocols on Mucosal OECs

With our previous protocol, ONFs proliferated at a higher rate than OECs, expanding their population over OECs. There have been reported successful methods to purify the culture from olfactory mucosal tissue: the differential adhesion technique to remove ONFs, growth factor supplement to stimulate OEC proliferation [57,58], and cell sorting devices. We implemented these methods and found that the differential adhesion approach worked well with bOEC cultures, as reported [58], but did not do well with mOECs. With the differential adhesion method, some OECs were lost through the process. We have also tested supplements in the culture medium with growth factors, such as NT3 [57], and found the morphology of OECs changed, suggesting possible changes in cell properties and reducing their reparative capacity. FACS helped to obtain a purer culture of mostly OECs, but the procedure could result in the loss of OECs from the already low number due to the variable expression level of OEC identifier P75. Purified OECs were cultured singularly and still proliferated at a low rate, and their differentiation seemed stunted, possibly due to the depletion of supportive factors from other cell types present in the original culture.

With the protocol used in this study, we have increased the OEC proportion in the mucosal cultures up to 20% and maintained normal OEC morphology. Forskolin affects cell mitosis and promotes differentiation by raising the cAMP level in the cell; N2 supplement is widely used in culture to help cell differentiation [59,60,61]. The combined effect of these two added factors in our culture leads to the suppression of ONF proliferation and the promotion of OEC differentiation. The decline of ONF proliferation improves the competitiveness of OECs for supportive factors, such as nutrients in the medium. The increased number of OECs in the culture also improves their paracrine support. Since N2 is a chemically defined supplement and forskolin has been shown safe to use in medicine [62,63], they are good candidates to prepare mMOECs for future clinical application.

### 4.3. Effect of Mucosal OECs Population

In our previous study, the transplantation of 5% OECs from the mucosal culture in the same rhizotomy model showed no regeneration of axons nor restoration of functions [31]. Some clinical trials have shown that, although safe, the transplantation of cell preparation from mucosa culture without any intervention or pieces of olfactory mucosa tissue resulted in little to no significant improvement of neurological functions [28,29], which could be due to the small number of OECs in the transplants. Our present study using 20% high-yield mMOECs showed significant improvement in the climbing performance test which we have never observed in the previous studies with 5% OECs. We also observed functional recovery in our newly adapted adhesive removal test. The results suggest that the higher population mOECs in the present culture preparation contribute to the restoration of the connections for touch and proprioception sensations.

In the present study, the deficits of function in two tests persisted with no improvement in the rats without the transplant over the experimental period. The OEC-ensheathed NF-positive axons in the reconstructed dorsal root entry zone and the BDA-traced dorsal root axons in the spinal cord, as shown in Figure 6, suggest that the functional restoration is associated with the regenerating axons growing into the spinal cord and the resumption of signal transmission. Presumably, the axons also formed connections in the ventral horn once they entered the spinal cord [64]. This observation is similar to our previous study of the transplantation of bulbar OECs in the same rhizotomy model, which demonstrated anatomical and electrophysiological regeneration of the dorsal root axons [16]. In our view, the unique property of OECs interacting with astrocytes provides a permissive pathway at the entry zone, allowing the regenerating axons to grow into the spinal cord [16,65,66]. To achieve this, a proportion of OECs higher than at least 5% is required. A future study with electrophysiological recording, e.g., [14,16], and improved axon tracing methods, e.g., using an adeno-associated virus, will provide more conclusive evidence.

### 4.4. Advantage of Using Biomaterial

Damage to the spinal cord often involves the progressive loss of tissue over time, resulting in cavitation at the injury site [67]. However, the limited mass of biopsy tissue yields far lower cell numbers than that needed for repair. Cell transplantation in the treatment of spinal cord and other CNS injuries are mainly implemented by microinjection at multiple locations and depths, which is time consuming and may cause damage to the host tissue. It is also difficult to retain the injected cells at the targeted site. 

In our previous studies, we successfully transplanted OEC aggregates formed by their own matrices [16,17,40]. Compared to microinjection of a cell suspension, the aggregates could be readily placed and reliably retained at the sites between the severed roots and the spinal cord. However, these aggregates required large numbers of densely packed cells. It is achievable in experimental settings but would be unattainable in a clinical application. The present study used collagen gel as a scaffold to encapsulate mMOECs and produced promising results. It allowed smaller numbers of OECs to distribute within a much larger volume and gave the cells structural support to expand and form a cellular network before transplantation. The collagen gel/cells formed a firm substrate, easily handled and trimmed into the required sizes. In our previous studies, when the human bulbar OECs (<10%) encapsulated in collagen were transplanted in the same rhizotomy model, only partial functional recovery was achieved [32], which is in marked contrast to a near full recovery with the transplantation of the rat bulbar OEC aggregates formed by their own matrices consisting of around 50% [16]. Collagen has been reported to promote neuroregeneration in peripheral nervous system injuries [68,69] and to support CNS axon sprouting by providing mechanical support and a favourable microenvironment when it is applied to fill in a lesion cavity formed by tissue damage after a brain or spinal cord injury [70,71,72]. Our previous and present studies suggest that the number of OECs in the transplant plays a major part in determining the extent of recovery, although the effect of collagen cannot be completely excluded. It has also been shown that collagen when transplanted in its solid state is impermissive to injured CNS axons [73]. The present study shows, however, that the preassembled gel of high-yield mOECs and collagen is effective in functional recovery. In future studies, the role of collagen alone needs to be investigated.

## 5. Conclusions

The transplantation of cultured olfactory ensheathing cells (OECs) from the olfactory bulb has great potential for repairing CNS injuries. However, a less invasive endonasal approach to the olfactory mucosa would be more favourable for obtaining OECs in clinical situations. The challenge has been that only a low yield of OECs can be harvested from the olfactory mucosa. The small number of mucosal OECs has less reparative efficacy. We have now developed a novel method that allows us to increase the mucosal OEC population up to 20%, compared with 5% from our previous protocol. Transplantation of these high-yield mOECs in our rhizotomised experimental model induced regeneration of dorsal root axons and restored function deficits, which we were unable to achieve with the population of 5% OECs in the transplant. Although the dorsal rhizotomy used in the present study does not reflect the larger, more variable damage encountered in the clinical relevance of human spinal plexus avulsions or spinal cord injuries, the finding does provide proof of principle with future development leading to clinical application. Our next step is to test our protocol on human material and explore whether the OEC population can be further increased without attenuating the properties of OECs in supporting neural regeneration.

## Figures and Tables

**Figure 1 cells-10-01186-f001:**
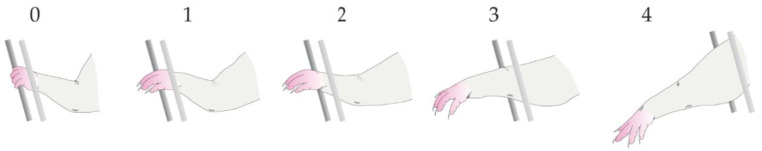
Error grading scheme for the degree of accuracy in the forepaw locating and grasping the grid bars. 0, successful grasp; 1, paw reaches grid-level but no grasping; 3, protrudes through the grid as far as the wrist; 3, as far as the elbow; 4, as far as the axilla.

**Figure 2 cells-10-01186-f002:**
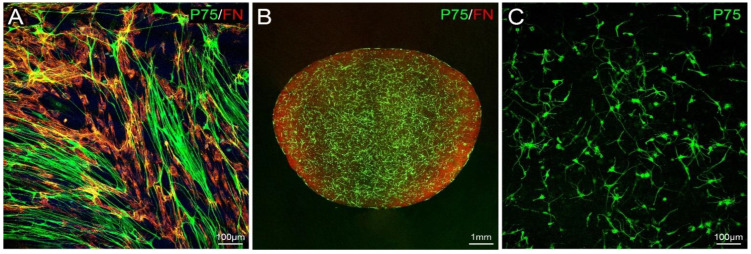
Immunohistochemistry for in vitro and in vivo studies. (**A**) A representative image of double immunofluorescence staining of mMOECs in a 21-day culture: OECs (P75, green), ONFs (FN, red); (**B**) an example of mMOEC encapsulation in collagen: OECs (P75, green), ONFs (FN, red); (**C**) higher power image to show the morphology of mMOECs (P75, green) and their distribution in collagen. Confocal images; scale bars: A, C, 100 µm; B, 1 mm.

**Figure 3 cells-10-01186-f003:**
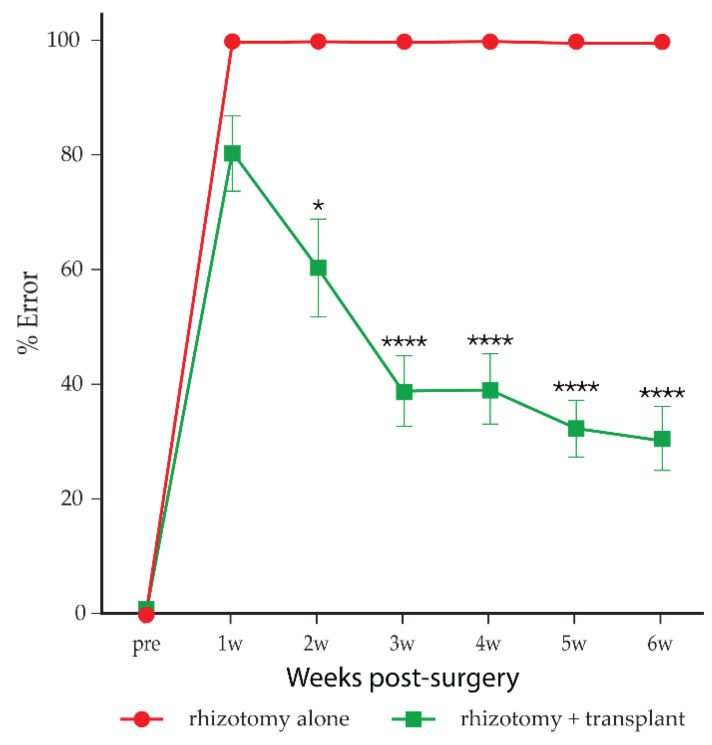
Mean error scores on a vertical climbing task. The rats in the control group that had rhizotomy alone (*n* = 6; red circles) did not show any noticeable functional improvement. The rats that had rhizotomy and received mMOEC transplants (*n* = 21; green squares) recovered progressively over 6 weeks of the testing period. Error bars: standard error of the mean. Two-way ANOVA, * *p* < 0.05, **** *p* < 0.0001 post hoc Sidak’s. Pre, pre-surgery; w, weeks.

**Figure 4 cells-10-01186-f004:**
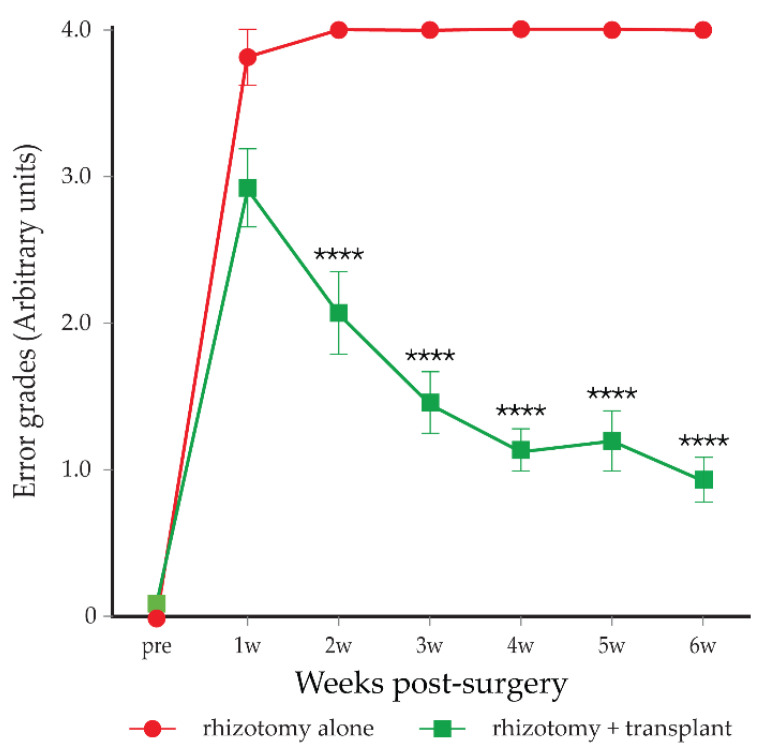
Mean forepaw fault score on the vertical climbing task—the magnitude of error in misplaced grasps. Red circles, rhizotomised rats (*n* = 6); green squares, rhizotomised rats with mMOEC transplants (*n* = 21). Scores over 6-week testing period. Mean ± standard error. Two-way ANOVA, **** *p* < 0.0001 post hoc Sidak’s.

**Figure 5 cells-10-01186-f005:**
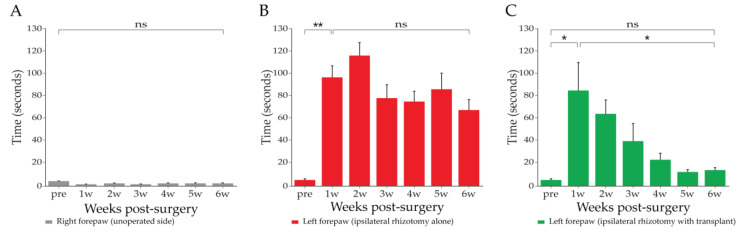
Adhesive removal test—mean time taken to notice and remove the sticker. (**A**) Right forepaw (on the unoperated side; *n* = 14), time taken to notice and remove the sticker was unaffected; (**B**) left forepaw (on the operated side with rhizotomy alone; *n* = 6), over 6-week testing period lengthened time taken remained unrecovered; (**C**) left forepaw (on the operated side with rhizotomy and transplant; *n* = 8), time taken recovered from initially around 80 s down to around 10 s. Bars: mean ± standard error. Two-way ANOVA, * *p* < 0.05, ** *p* < 0.005 post hoc Sidak’s.

**Figure 6 cells-10-01186-f006:**
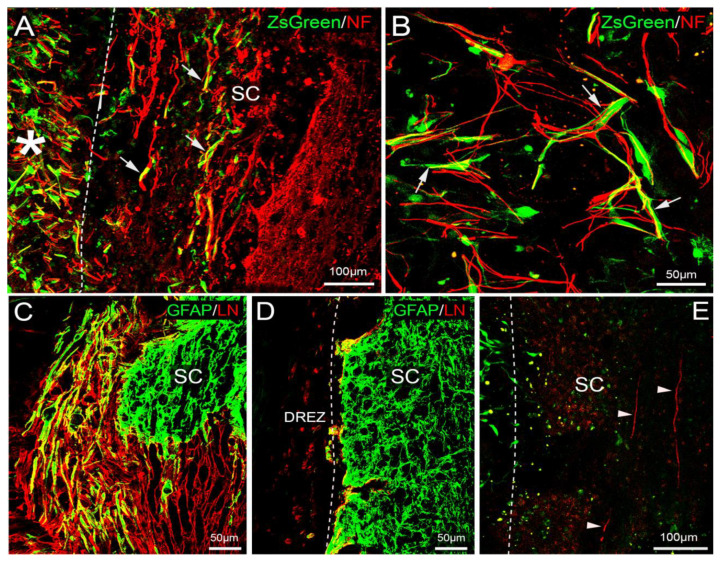
(**A**) NF antibody immunostained axons (red) with transplanted cells expressing ZsGreen (green) in the dorsal root entry zone. Transplanted mMOECs ensheathed some NF+ axons. Some transplanted cells migrated into the spinal cord (SC) and ran parallel to the host axons (arrows); (**B**) high power view of transplanted mMOECs (green) ensheathing NF+ regeneration axons (red) at the dorsal root entry zone (arrows); (**C**) double immunofluorescence staining of GFAP (green) and LN (red) showing the interaction between PNS and CNS tissue. The astrocytic processes extended out into the reapposed DR; (**D**) the astrocytic processes (green) did not extend into the dorsal root entry zone (DREZ) in the rhizotomy alone group; (**E**) BDA labelled dorsal root axons (red) in the spinal cords (arrowheads, SC). Survival time after transplantation: six weeks. Cryostat horizontal sections; confocal images.

## Data Availability

Not applicable.

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
