# Peer review of "High-Yield Mucosal Olfactory Ensheathing Cells Restore Loss of Function in Rat Dorsal Root Injury"

_cells, 2021, doi:10.3390/cells10051186_

Round 1
Reviewer 1 Report
This is a carelessly written paper with inadequate and incomplete descriptions of the methods, the behavioural data. There are numerous lingustic and punctuation errors and the results section can be much better illustrated e.g. there are no illustrations of BDA labelled regenerating axons and it is impossible to tell if recovery is really due to proprioceptors, since no physiological data is presented. Too many abbreviations are used and a majority are not explained in the text. The discussion does not address the results presented nor the limitations of this work. It needs an entire rewrite.
My comments and observations are attached in the pdf.
My comments are attached

Author Response
Responses to the Reviewer 1:
We are grateful for the comments and suggestions. We have dealt with all the points and believe that the MS has been improved.
This is a carelessly written paper with inadequate andincomplete descriptions of the methods, the behavioural data.There are numerous lingustic and punctuation errors and theresults section can be much better illustrated e.g. there are noillustrations of BDA labelled regenerating axons and it isimpossible to tell if recovery is really due to proprioceptors, sinceno physiological data is presented. Too many abbreviations areused and a majority are not explained in the text. The discussiondoes not address the results presented nor the limitations of thiswork. It needs an entire rewrite.
We have given more details of the methods and behavioural data in the Materials and Methods section. A new figure is added to explain the scoring scheme for the vertical climbing test. The linguistic and punctuation errors have been corrected. Images of BDA labelling for the axons, and GFAP/FN staining for the control (in new Figure 6 ) are added. We have checked and defined all the abbreviations. The Discussion section has been condensed, shortened and some parts are re-written.
Line 48: ‘s’ is deleted
Line 57: Changed ‘climb’ to ‘climbing’ and added ‘of’.
Line 67: Added sex and number of the rats.
Line 70: Number of rats is added.
Line 85: % is added; abbreviation of ITS defined.
Line 86: Abbreviation of PS is already defined on line 74.
Line 93: ‘DMED’ is corrected.
Line 96: Abbreviation of mMOECs is defined.
Line 98: Abbreviation of TE is already defined.
Line 116: Changed ‘closely’ to ‘close’.
Line 124: Number of rats for BDA labelling is corrected
Line 128: A sentence of ‘4 weeks after transplantation. For the controls, 4 weeks after rhizotomy’ is added.
Line 139: A new sentence is added – ‘Autotomy was observed in 2 animals. If the autotomy occurred to the extent of involving two joints in one toe, or past the nail, in two toes, the animals were culled humanely.’
Line 143: The assessments were carried out by two people blindly to experimental groups. A new figure is added to illustrate the scoring scheme.
Line 156: This test was developed later than the climbing test, 6 from the control group and 8 from the transplanted group among the 21 rats were used for this test.
Line 163: A full stop is added after ‘recorded’.
Line 179: The concentration is given. This reagent is ready-made for use by the manufacturer.
Line 190: Abbreviation of ONF is defined.
Line 185: Changed the font to italic.
Line 188: Unbolded. The word is not bolded in the submitted MS.
Line 190: ‘avulsed’ deleted.
Line 192: More details are added to the text – ‘The tissue block was mounted with an embedding compound Cryo-M-Bed (Fisher Scientific, UK) on a cutting chuck and frozen in the Cryostat chamber (Leica CM3050) at -20 degrees until hardened. Serial sections of 16µm horizontal plane at cervical levels C6-T1 were cut and thaw-mounted onto slides.’
Line 204: Living Colors® Anti-RCFP Polyclonal Pan Antibody is a product for Takare Bio.
Line 208: A sentence of ‘The laser intensity, pinhole sizes and sensitivity of the photomultipliers were adjusted accordingly to ensure no bleed through to cross channels.’ added to the text.
Line 216: Definition of the abbreviation is given.
Line 225-226: The sentence is re-written – ‘As the culture progresses the dominating expansion of ONF's led to the morphology of the OECs to change into large and flat shapes with irregular shapes and short processes appearing senescent.’ The missing bracket is inserted.
Line 241: Changed back ‘in vitro and in vivo’ to Italics. Italics were applied in the submitted MS.
Line 244: Quantification of time taken has been added to the text – ‘The average time taken to complete one climb ranged from 10-30 seconds. Sometimes a rat was reluctant to climb or it can stop in the middle of the climb. We did not take this as a measure because the time taken was so variable.’
Line 249, 254: Description was given in ‘2.5.1. Vertical climbing test’. A new figure is now added to explain the scoring scheme.
Line 256: Agreed, we have rephrased the sentence.
Line 268: Figure 2, % error related to grades 1-4 has been explained with more details in the M&M 2.5.1. Section. % error scores any miss of grasping. Grades 1-4 grades the increasing order of severity of the deficit.
Line 274: The title is changed to ‘Forepaw Fault Score ’. Please see the explanation on Line 268.
Line 292: Foot fault is deleted as suggested.
Figure 3 (new figure 4): Thanks for pointing out the error. It is corrected.
Line 300: The numbers are given.
Line 305: This was the test that we developed later than the climbing test, 6 from the control group and 8 from the transplanted group among the 21 rats were used for this test. Mean ± SEMs have been provided.
Line 321 and Figure 2: The error bars in each group are added to the figure; Corrected the spelling mistake of ‘rhizotomy’.
Line 322: Numbers per group are added to the legend.
Line 334-337: We have labelled Figure 5A, B (New Figure 6) with asterisk and arrows to indicate cell location. The sentences are re-written.
Line 342: We have added a dotted line to Figure 5A (New figure 6) to show the boundaries of DREZ and the spinal cord. Labelled the migrated cells with arrows in Figure 5 A, B (New Figure 6).
Line 346: Our mistake, we have removed D-F.
Line 354-358: A image of BDA labelling is added in Figure 5 (New Figure 6E).
Line 366: Added a picture in the new Figure 6 to show no outgrowth of the astrocytic processes that cross the boundary in the rhizotomy alone control group.
Line 378: Added a new paragraph about the results, the strengths and weakness of the methods used, and the clinical relevance of using rhizotomy vs true avulsion of the root at the beginning of the Discussion.
Line 379: We have shortened the section.
Line 461: We have rephrased the sentence.
We agree that other potential mechanisms such as pattern generators or intraspinal reorganization could be also contributed to the recovery of the functions after injuries to the spinal cord. In the present study, the deficits persisted in our rhizotomy alone group over the six weeks testing period but were recovered in the transplanted group. The recovery by potential mechanisms such as pattern generators or intraspinal reorganization may not interpret the outcome in this study with the rhizotomy model.
Reviewer 2 Report
line 150-152: was the scoring performed by blinded observer or was it done using a software? Please add information on how was the scoring of the animals was performed. In case the scores were done by a person, was the person performing the scoring unaware of the treatment groups? In case the scoring was performed by software , add details of the software. Include details on the scoring parameters and how exactly the scoring was done. This is important information that will help readers interpret the outcomes.
line 155: was the test performed always on the same paw first? Did the authors take any precautions to prevent conditioning of the animals? Animals in behavioural experiments often become conditioned to the test. It is important to give details about if and how the conditioning of animals was avoided. Please also add details if the test was performed on the right paw first or if the authors assessed the paws randomly. E.g. if always the right paw was assessed first, the measurement of the left paw maybe influenced by the stress caused by the first measurement. This is an important information for the readers when interpreting the outcomes.
line 191: please add details of the freezing process: e.g. what freezing media was used, temperature, method.
line 217: please define ONFs
results:
all experiments are missing important control groups that would show that the effects seen in those experiments are specific to the rhizotomy model and /or to the treatments:
- It would be important to see the effect of a sham surgery group, as we can
not exclude that the effects that the authors are assessing are not due to
inflammatory processes associated with tissue damage itself.
The authors should asses the effects of the surgery without actually
cutting the nerves (sham surgery group). If those data was already
published previously by the authors, then please add references. - The authors should exclude that the collagen solution alone (vehicle)
has no effect on the outcomes of the study by performing rhizotomy and
the effects can be attributed to the mMOEC transplants by administering
vehicle (collagen solution without cells) to the animals and assessing
the efects.
In my opinion, to be able to prove the positive effects of mMOEC transplants on rhizotomy the authors should compare the results with those negative controls that will assure that the effect can be attributed to the transplanted cells and exclude other non-neuronal factors such as inflammation caused by tissue damage following the surgery or the effects of collagen alone.
Author Response
Responses to the Reviewer 2:
We are grateful for the comments and suggestions. We have dealt with the points and believe that the MS has been improved.
line 150-152: was the scoring performed by blinded observer or was it done using a software? Please add information on how was the scoring of the animals was performed. In case the scores were done by a person, was the person performing the scoring unaware of the treatment groups? In case the scoring was performed by software, add details of the software. Include details on the scoring parameters and how exactly the scoring was done. This is important information that will help readers interpret the outcomes.
Yes, all the scoring were carried out by two people blindly to experimental groups.We have added a figure (new Figure 1) to explain the scoring scheme.
line 155: was the test performed always on the same paw first? Did the authors take any precautions to prevent conditioning of the animals? Animals in behavioural experiments often become conditioned to the test. It is important to give details about if and how the conditioning of animals was avoided. Please also adddetails if the test was performed on the right paw first or if the authors assessed the paws randomly. E.g. if always the right paw was assessed first, the measurement of the left paw maybe influenced by the stress caused by the first measurement. This is an important information for the readers when interpreting the outcomes.
Line 155: We agreed, animals in behavioural experiments often become conditioned to the test. In this function test, the first paw tested between the left and right side was randomised to prevent conditioning. We added randomisation of the testing order of the left and right paws in the M&M. However, the deficit persistently unimproved in the rhizotomy alone group over the six-week testing period.
line 191: please add details of the freezing process: e.g. what freezing media was used, temperature, method.
Line 191: More details of the freezing process including freezing media, temperature, and method were added in the text – ‘The tissue block was mounted with an embedding compound Cryo-M-Bed (Fisher Scientific, UK) on a cutting chuck and frozen in the Cryostat chamber (Leica CM3050) at -20 degrees until hardened. Serial sections of 16µm horizontal plane at cervical levels C6-T1 were cut and thaw-mounted onto slides’
Line 217: Abbreviation of ONFs is defined.
Results:
all experiments are missing important control groups that wouldshow that the effects seen in those experiments are specific tothe rhizotomy model and /or to the treatments:
It would be important to see the effect of a sham surgery group,as we can not exclude that the effects that the authors are assessing arenot due to inflammatory processes associated with tissue damage itself. The authors should asses the effects of the surgery withoutactually cutting the nerves (sham surgery group). If those data wasalready published previously by the authors, then please add references.
We have used this model for a series of studies. In our previous pilot study, we did perform a group of sham surgery by exposing unilateral 4 dorsal roots and the spinal cord, but without cutting the nerves. We assessed the climbing and adhesive-removal tests and did not observe any behavioural change or loss of function. We have now included this sham surgery in the text - ‘In a pilot study, four rats received a sham surgery performance by exposing the four unilateral dorsal roots and the spinal cord, but without cutting the nerves.’ in the section of M&M; ‘The rats with the sham surgery did not show any abnormal behaviour and functional deficits in this test after they recovered to their normal cage activities after the surgery.’ In the section of Results.
The authors should exclude that the collagen solution alone(vehicle) has no effect on the outcomes of the study by performingrhizotomy and the effects can be attributed to the mMOEC transplants byadministering vehicle (collagen solution without cells) to the animals andassessing the efects.
In the present study, we did not transplant collagen alone in the rhizotomy model, we cannot fully exclude the possibility that the collagen alone might affect the outcomes. However, in our previous study, we transplanted human bulbar OECs in collagen gel only produced partial functional recovery compared with the present study [1]. We appreciate the reviewer’s suggestion and will do an experiment by transplantation of the collagen gel alone in our future studies.
In my opinion, to be able to prove the positive effects of mMOEC transplants on rhizotomy the authors should compare the results with those negative controls that will assure that the effect can be attributed to the transplanted cells and exclude other non-neuronal factors such as inflammation caused by tissue damage following the surgery or the effects of collagen alone
We thank the reviewer for pointing this out. There are indeed a number of factors that contribute to the OECs’ property in neural repair [2-4]. The main focus of the present study is on comparison of the improved 20% mOEC population vs 5%. We only examined axonal regeneration and functional restoration in this study, so we could not exclude other non-neuronal factors such as inflammation caused by tissue damage following the surgery. It is a very interesting and important issue that the reviewer raised, we will consider them in our future study.
- Collins, A.; Li, D.; Liadi, M.; Tabakow, P.; Fortuna, W.; Raisman, G.; Li, Y. Partial Recovery of Proprioception in Rats with Dorsal Root Injury after Human Olfactory Bulb Cell Transplantation. J Neurotrauma 2018, 35, 1367-1378, doi:10.1089/neu.2017.5273.
- Pastrana, E.; Moreno-Flores, M.T.; Gurzov, E.N.; Avila, J.; Wandosell, F.; Diaz-Nido, J. Genes associated with adult axon regeneration promoted by olfactory ensheathing cells: a new role for matrix metalloproteinase 2. jnsci 2006, 26, 5347-5359.
- Franssen, E.H.; De Bree, F.M.; Essing, A.H.; Ramon-Cueto, A.; Verhaagen, J. Comparative gene expression profiling of olfactory ensheathing glia and Schwann cells indicates distinct tissue repair characteristics of olfactory ensheathing glia. Glia 2008, 56, 1285-1298, doi:10.1002/glia.20697.
- Chuah, M.I.; Hale, D.M.; West, A.K. Interaction of olfactory ensheathing cells with other cell types in vitro and after transplantation: glial scars and inflammation. Exp Neurol 2011, 229, 46-53, doi:10.1016/j.expneurol.2010.08.012.
Round 2
Reviewer 1 Report
This is a substantially improved revision. The authors have addressed the vast majority of my concerns. However, the discussion still does not really address how functional recovery was attributable to proprioceptive regeneration and the single BDA image does not provide any conclusive evidence of connections in the ventral horn. Electrophysiological recordings would have provided conclusive evidence (one way or the other). There is still no discussion as to potential mechanisms of touch and proprioceptive recovery, when it occurs or by what methods e.g. reestablished circuits or newly generated circuits. According to the graphs touch is only significantly recovered at 6 weeks whereas behaviour is by 4 weeks. Some discussion of this is necessary and is of interest to those in the field.
Minor other grammatical and linguistic comments are provided in the attached pdf.

Author Response
Many thanks for the further comments and suggestions.
Responses to Reviewer:
Comments and Suggestions for Authors
This is a substantially improved revision. The authors have addressed the vast majority of my concerns.
However, the discussion still does not really address how functional recovery was attributable to proprioceptive regeneration and the single BDA image does not provide any conclusive evidence of connections in the ventral horn. Electrophysiological recordings would have provided conclusive evidence (one way or the other).
We have added a part in the DISCUSSION to address the reviewer’s comment,
We agree with the reviewer’s point that the BDA image does not provide conclusive evidence of connections in the ventral horn. In the present study, we only prepared a horizontal plane of sections for BDA labelling. We aimed to identify the dorsal root axons that cross the entry zone into the spinal cord. In future studies, we will improve the sensitivity of our tracing method and investigate the termination of the regenerated axons in coronal sections.
There is still no discussion as to potential mechanisms of touch and proprioceptive recovery, when it occurs or by what methods e.g. reestablished circuits or newly generated circuits.
We have added a section in Discussion 4.3:
‘In the present study, the deficits of function in two tests persisted with no improvement over the experimental period in the rats without the transplant. The OEC-ensheathed NF-positive axons in the reconstructed dorsal root entry zone and the BDA-traced dorsal root axons in the spinal cord as shown in Figure 6 suggest the functional restoration is associated with the regenerating axons growing into the spinal cord and resumption of signal transmission. Presumably, the axons also formed connections in the ventral horn once they have entered the spinal cord [1]. This observation is similar to our previous study by transplantation of bulbar OECs in the same rhizotomy model which demonstrated anatomically and electrophysiologically regeneration of the dorsal root axons [2]. In our view, the unique property of OECs interacting with astrocytes provides a permissive pathway at the entry zone allowing the regenerating axons to grow into the spinal cord [2-4]. To achieve this a proportion of OECs higher than at least 5% is required. A future study with electrophysiological recording e.g. [2,5] and improved axon tracing methods e.g. using adeno-associated virus will provide more conclusive evidence.’
According to the graphs touch is only significantly recovered at 6 weeks whereas behaviour is by 4 weeks. Some discussion of this is necessary and is of interest to those in the field.
Figure 5 and the text are amended to give more details of the recovery data.
A sentence of ‘The recovering rates of the two functional tests appear to be similar’ is added to the first paragraph of the discussion.
Line 112: reworded to ‘Unilateral transection of four dorsal roots’.
Line 144: ‘… one animal in the control and one in the transplanted group.’ is added to the text.
Line 163: ‘… and the scores from the two assessors were averaged’ is added to the text.
Line 205: The word was not bolded in the original text. It is somehow changed by the Journal. It is un-bolded again.
Line 309: (see Figure 1) is added to the text.
Line 391-394: The sentence is removed and integrated with the Discussion.
- Massey, J.M.; Amps, J.; Viapiano, M.S.; Matthews, R.T.; Wagoner, M.R.; Whitaker, C.M.; Alilain, W.; Yonkof, A.L.; Khalyfa, A.; Cooper, N.G.; et al. Increased chondroitin sulfate proteoglycan expression in denervated brainstem targets following spinal cord injury creates a barrier to axonal regeneration overcome by chondroitinase ABC and neurotrophin-3. Experimental Neurology 2008, 209, 426-445.
- Ibrahim, A.G.; Kirkwood, P.A.; Raisman, G.; Li, Y. Restoration of hand function in a rat model of repair of brachial plexus injury. Brain 2009, 132, 1268-1276, doi:10.1093/brain/awp030.
- Li, Y.; Carlstedt, T.; Berthold, C.H.; Raisman, G. Interaction of transplanted olfactory-ensheathing cells and host astrocytic processes provides a bridge for axons to regenerate across the dorsal root entry zone. Experimental Neurology 2004, 188, 300-308, doi:10.1016/j.expneurol.2004.04.021.
- Lakatos, A.; Franklin, R.J.M.; Barnett, S.C. Olfactory ensheathing cells and Schwann cells differ in their in vitro interactions with astrocytes. Glia 2000, 32, 214-225.
- Toft, A.; Scott, D.T.; Barnett, S.C.; Riddell, J.S. Electrophysiological evidence that olfactory cell transplants improve function after spinal cord injury. Brain 2007, 130, 970-984.
Reviewer 2 Report
Regarding the point below: I believe that the addition of experiments performed with the appropriate negative controls would greatly improve the quality of the manuscript. If the authors are unable to perform experiments with collagen controls, then this weakness of the study should be addressed and discussed in the discussion section of the manuscript.
The authors should exclude that the collagen solution alone(vehicle) has no effect on the outcomes of the study by performing rhizotomy and the effects can be attributed to the mMOEC transplants by administering vehicle (collagen solution without cells) to the animals and assessing the effects.
In the present study, we did not transplant collagen alone in the rhizotomy model, we cannot fully exclude the possibility that the collagen alone might affect the outcomes. However, in our previous study, we transplanted human bulbar OECs in collagen gel only produced partial functional recovery compared with the present study [1]. We appreciate the reviewer’s suggestion and will do an experiment by transplantation of the collagen gel alone in our future studies
Author Response
Many thanks for the further comments and suggestions.
Responses to Reviewer 2:
Comments and Suggestions for Authors
Regarding the point below: I believe that the addition of experiments performed with the appropriate negative controls would greatly improve the quality of the manuscript. If the authors are unable to perform experiments with collagen controls, then this weakness of the study should be addressed and discussed in the discussion section of the manuscript.
We agreed with the reviewer and have added a section on the role of collagen in the Discussion and added some related references.
“In our previous studies when the human bulbar OECs (<10%) encapsulated in collagen were transplanted in the same rhizotomy model only partial functional recovery was achieved [1], which is in marked contrast to a near full recovery with transplantation of the rat bulbar OEC aggregates formed by their own matrices consisting of around 50% [2]. Collagen has been reported to promote neuroregeneration in the peripheral nervous system injuries [3,4] and to support CNS axon sprouting by providing mechanical support and a favourable microenvironment when it is applied to fill in a lesion cavity formed by tissue damage after a brain or spinal cord injury [5-7]. Our previous and present studies suggest that the number of OECs in the transplant plays a major part in determining the extent of recovery, although the effect of collagen cannot be completely excluded. It has also been shown that collagen when transplanted in its solid state is impermissive to injured CNS axons [8]. The present study shows however the pre-assembled gel of high-yield-mOECs and collagen is effective in functional recovery. In future studies, the role of collagen alone needs to be investigated.”
- Collins, A.; Li, D.; Liadi, M.; Tabakow, P.; Fortuna, W.; Raisman, G.; Li, Y. Partial Recovery of Proprioception in Rats with Dorsal Root Injury after Human Olfactory Bulb Cell Transplantation. J Neurotrauma 2018, 35, 1367-1378, doi:10.1089/neu.2017.5273.
- Ibrahim, A.G.; Kirkwood, P.A.; Raisman, G.; Li, Y. Restoration of hand function in a rat model of repair of brachial plexus injury. Brain 2009, 132, 1268-1276, doi:10.1093/brain/awp030.
- Taras, J.S.; Jacoby, S.M.; Lincoski, C.J. Reconstruction of digital nerves with collagen conduits. J Hand Surg Am 2011, 36, 1441-1446, doi:10.1016/j.jhsa.2011.06.009.
- Khaing, Z.Z.; Schmidt, C.E. Advances in natural biomaterials for nerve tissue repair. Neurosci Lett 2012, 519, 103-114, doi:10.1016/j.neulet.2012.02.027.
- Zhong, Y.; Bellamkonda, R.V. Biomaterials for the central nervous system. J R Soc Interface 2008, 5, 957-975, doi:10.1098/rsif.2008.0071.
- Joosten, E.A.; Bar, P.R.; Gispen, W.H. Collagen implants and cortico-spinal axonal growth after mid-thoracic spinal cord lesion in the adult rat. J Neurosci Res 1995, 41, 481-490, doi:10.1002/jnr.490410407.
- Orive, G.; Anitua, E.; Pedraz, J.L.; Emerich, D.F. Biomaterials for promoting brain protection, repair and regeneration. Nature reviews. Neuroscience 2009, 10, 682-692, doi:10.1038/nrn2685.
- Joosten, E.A.J.; B„r, P.R.; Gispen, W.H. Directional regrowth of lesioned corticospinal tract axons in adult rat spinal cord. Neuroscience 1995, 69, 619-626.